# ZIP8-Mediated Intestinal Dysbiosis Impairs Pulmonary Host Defense against Bacterial Pneumonia

**DOI:** 10.3390/ijms23031022

**Published:** 2022-01-18

**Authors:** Derrick R. Samuelson, Deandra R. Smith, Kelly C. Cunningham, Todd A. Wyatt, Sannette C. Hall, Daryl J. Murry, Yashpal S. Chhonker, Daren L. Knoell

**Affiliations:** 1Department of Internal Medicine-Pulmonary Division, College of Medicine, University of Nebraska Medical Center, Omaha, NE 68198, USA; kelly.cunningham@unmc.edu (K.C.C.); twyatt@unmc.edu (T.A.W.); 2Department of Pharmacy Practice and Science, College of Pharmacy, University of Nebraska Medical Center, Omaha, NE 68198, USA; deandra.smith@unmc.edu (D.R.S.); sannette.hall@unmc.edu (S.C.H.); dj.murry@unmc.edu (D.J.M.); y.chhonker@unmc.edu (Y.S.C.); 3Department of Environmental, Agricultural and Occupational Health, College of Public Health, University of Nebraska Medical Center, Omaha, NE 68198, USA; 4Veterans Affairs Nebraska-Western Iowa Health Care System, Omaha, NE 68105, USA

**Keywords:** zinc, pneumonia, microbiome, zinc transporter, host defense, gut-lung axis

## Abstract

Pneumococcal pneumonia is a leading cause of morbidity and mortality worldwide. An increased susceptibility is due, in part, to compromised immune function. Zinc is required for proper immune function, and an insufficient dietary intake increases the risk of pneumonia. Our group was the first to reveal that the Zn transporter, ZIP8, is required for host defense. Furthermore, the gut microbiota that is essential for lung immunity is adversely impacted by a commonly occurring defective ZIP8 allele in humans. Taken together, we hypothesized that loss of the ZIP8 function would lead to intestinal dysbiosis and impaired host defense against pneumonia. To test this, we utilized a novel myeloid-specific *Zip8*KO mouse model in our studies. The comparison of the cecal microbial composition of wild-type and *Zip8*KO mice revealed significant differences in microbial community structure. Most strikingly, upon a *S. pneumoniae* lung infection, mice recolonized with *Zip8*KO-derived microbiota exhibited an increase in weight loss, bacterial dissemination, and lung inflammation compared to mice recolonized with WT microbiota. For the first time, we reveal the critical role of myeloid-specific ZIP8 on the maintenance of the gut microbiome structure, and that loss of ZIP8 leads to intestinal dysbiosis and impaired host defense in the lung. Given the high incidence of dietary Zn deficiency and the ZIP8 variant allele in the human population, additional investigation is warranted to improve surveillance and treatment strategies.

## 1. Introduction

Community-acquired pneumonia (CAP) is a leading cause of morbidity and mortality worldwide. *Streptococcus pneumoniae* (pneumococcus) remains the most commonly identified cause of CAP in the US [1,2,3,4]. The incidence of CAP continues to rise, contributing to increased hospitalizations and mortality [5,6]. A major cause of CAP is compromised immune function [7]. Zinc (Zn) is required for proper immune function [8,9,10], and an insufficient dietary intake is highly prevalent within vulnerable populations [11,12]. Zn-deficient subjects are more susceptible to bacterial and viral pathogens [13] and have a higher incidence of pneumonia [14,15,16] while Zn supplementation can reduce the incidence of pneumonia [17,18,19].

Accumulating evidence highlights the influence of the gut microbiota on lung immunity, referred to as the gut-lung axis. The human body is colonized by a vast number of microbes, with the gut being the most densely colonized organ. Immune homeostasis is dependent on a microbiome that provides microbial metabolites for the appropriate maturation and priming of the immune system [20]. In humans, environmental factors, such as diet, can shift the gut microbiota towards a decreased abundance of beneficial bacterial species accompanied by an outgrowth of pathogenic ones, referred to as dysbiosis. Importantly, intestinal dysbiosis negatively impacts the regulation of the gut-lung axis and is linked to an increased susceptibility to respiratory infections [21].

Zn homeostasis in mammals is tightly regulated by a constellation of 24 Zn transporters. Our group was the first to reveal that the Zn transporter, ZIP8, is unique, relative to other family members, in that it is required by myeloid-lineage cells to maintain proper host defense against bacterial pneumonia [22,23,24]. Recent human studies have revealed that a frequently occurring ZIP8 variant allele that leads to defective intracellular Zn transport (rs13107325; Ala391Thr risk allele) is strongly associated with inflammation-based disorders [25,26] and bacterial infection [27]. Based on this, our laboratory created a novel myeloid-specific ZIP8 knockout (*Zip8*KO) mouse model and determined whether ZIP8 loss would alter host protection against pneumococcal pneumonia via changes in the gut microbiota. Studies were conducted using a cecal adoptive transfer model with antibiotic-cleansed mice followed by *S. pneumoniae* instillation into the lung. First, we compared the microbiota of fecal samples obtained from WT and *Zip8*KO mice and observed substantial differences in microbial composition. The intestinal composition of butyrate and branched short chain fatty acids was also significantly altered in fecal samples from *Zip8*KO mice in a manner that would adversely impact immune function. Most strikingly, following the instillation of pneumococcus into the lung, we observed a significant increase in weight loss, inflammation, and lung tissue damage in mice previously transplanted with the microbiota from *Zip8*KO mice, when compared to mice that received microbiota from WT mice. These results, for the first time, reveal a novel and potentially vital axis that exists between Zn, the ZIP8-mediated alteration of the gut microbiome, and host defense against pneumococcus (and likely other pathogens). Based on these novel observations, we show that Zn dyshomeostasis, originating within the gut, causes the dysbiosis of the gut microbiome, adversely impacting immune function in the lung in response to pneumococcus.

## 2. Results

### 2.1. ZIP8 Loss Alters the Intestinal Microbial Community

We compared the composition of the intestinal microbial communities between wild-type (WT) and *Zip8*KO mice. The loss of ZIP8 expression in myeloid lineage cells did not significantly affect alpha (α)-diversity. Conversely, the beta (β)-diversity between WT and *Zip8*KO mice was affected by the loss of ZIP8 expression, as determined by a distance-based redundancy analysis (dbRDA) on the sample-wise Bray–Curtis dissimilarity distances using phyloseq and vegan. Significant differences in the β-diversity were observed between WT and *Zip8*KO mice (q < 0.05) (Figure 1A). We then determined the amplicon sequence variants (ASV) that exhibited a significantly differential abundance in the cecum at the genus levels using STAMP (Figure 1B). Specifically, we found that bacteria from the genuses *Desulfovibrio* (ASV16) and *Intestinimonas* (ASV65), as well as the families *Clostridiales Family*_*XIII* (AVS140) and *Lachnospiraceae* (ASV30), were enriched in WT mice compared to *Zip8*KO mice (Figure 1B). Bacteria from the genuses *Muribaculum* (ASV53 and 67), *Erysipelatoclostridium* (ASV129), *Mucispirillum* (ASV33), *Parasutterella* (ASV17), and Prevotellaceae_UCG-001 (ASV12), as well as from the family *Ruminococcaceae* (ASV115), were enriched in *Zip8*KO mice compared to WT mice (Figure 1B). We also performed a PICRUSt2 analysis to assess the predicted functional capacity of the intestinal microbiota. We found that, similar to β-diversity, the inferred functional capacity was significantly altered by the loss of ZIP8 expression in myeloid lineage cells, as determined by a principal coordinate analysis based off the inferred functional capacity (Figure 1C). We also evaluated the individual functional pathways of the intestinal microbiota and found that 14 predicted pathways were significantly different between *Zip8*KO mice and WT mice (Figure 1D). Notably, we observed a decrease in two predicted butyrate synthesis pathways in *Zip8*KO mice (acetyl-CoA fermentation to butanonate II and succinate fermentation to butanotate) compared to WT mice.

### 2.2. ZIP8 Loss Decreases Intestinal Butyrate Levels

Given the potential impact on butyrate production observed at both the taxa and inferred function levels, we sought to determine the intestinal levels of butyrate. Butyrate is a known bioactive microbial metabolite implemented in immune regulation [28]. Cecal levels of butyrate were significantly decreased in *Zip8*KO mice when compared to WT mice measured via LC-MS/MS (Figure 2A). The other common short-chain fatty acids, including propionic acid, acetic acid, and valeric acid, were not different between *Zip8*KO and WT mice (Figure 2B,C, respectively). However, the branched-SCFAs (BCFAs), 2-methylbutyric acid (Figure 2D), isobutyric acid (Figure 2E), and isovaleric acid (Figure 2F), were all increased in *Zip8*KO mice when compared to WT mice.

### 2.3. ZIP8-Mediated Dysbiosis Increases Susceptibility to Streptococcus pneumoniae in the Lung

To determine the role of ZIP8-mediated intestinal dysbiosis on pulmonary host defense against pneumococcal infection, naïve antibiotic-cleansed mice were recolonized with the microbiota from WT and *Zip8*KO mice and were allowed to establish recolonization stability. We first assessed the composition of the intestinal microbial communities between the donor and recipient *Zip8*KO mice and WT mice. Microbiota composition comparisons were assessed following one week of recolonization. The analysis of the β-diversity demonstrated that the microbial communities were not different between the *Zip8*KO donor and the *Zip8*KO-microbiota recipient mice (*p* = 0.1116), or between the WT microbiota donor and the WT microbiota recipient mice (*p* = 0.1127) (Figure 3). However, the distinction between *Zip8*KO and WT microbiota in the recolonized mice was preserved and strongly significantly different (*p* = 0.00018) (Figure 3). Recolonized mice were then administered, via intranasal installation, *S. pneumoniae* (4 × 10^8^ CFU) and were sacrificed 48 h post-infection. *Zip8*KO mice were observed to have an increased weight loss following pneumococcal infection in the lung (Figure 4A). The histological staining of lung tissue revealed that *Zip8*KO mice had increased lung injury (representative images, Figure 4B), as determined by their inflammatory score (Figure 4C) and increased cell death (representative images, Figure 4D), as determined by TUNEL staining (Figure 4E). To determine whether increased lung tissue damage altered bacterial dissemination, bacterial counts were enumerated from the lung and spleens of both groups. *Zip8*KO mice exhibited a significantly higher bacterial burden in both the lung (Figure 4F) and spleen (Figure 4G) 48 h post-infection.

### 2.4. ZIP8-Mediated Dysbiosis Increases Lung Inflammation Following Bacterial Instillation

To examine the immune response against *S. pneumoniae* in the lung, WT and *Zip8*KO mice were administered 4 × 10^8^ CFU *S. pneumoniae* and were euthanized 48 h later. The analysis of the bronchoalveolar lavage (BAL) fluid from the lungs revealed that the bacterial instillation resulted in increased total numbers of leukocytes in the airways of both groups when compared to the uninfected mice (Figure 5A). Neutrophils were the dominant subset of leukocytes and were significantly increased in *Zip8*KO mice compared to their WT counterparts (Figure 5B). This result also corresponded with an increase in the total proteins in the BAL fluid (Figure 5C). The *Zip8*KO mice exhibited a trend in higher numbers of macrophages (Figure 5D) and lymphocytes (Figure 5E) but this did not achieve statistical significance. The pneumococcal infection was also associated with a significant increase in the expression of IL-6, TNF-α, CXCL1, and IFNγ in both groups and was significantly higher in the *Zip8*KO group when compared to the WT animals (*p* = 0.055, Figure 6A–D). Taken together, these results demonstrate an imbalance in the cellular landscape, leading to an exaggerated immune response to *S. pneumoniae* as a consequence of ZIP8-mediated intestinal dysbiosis.

### 2.5. ZIP8-Mediated Dysbiosis Alters Lung Tissue Cellularity

Mice were infected with *S. pneumoniae* (4 × 10^8^ CFUs) and the immune cells in the lung homogenates from both groups were characterized using flow cytometry. After gating out debris and doublets, live CD45^+^ leukocytes were selected. Cells were characterized as AMs, based on the expression of CD11c, Siglec F, and CD64 (CD45^+^CD11c^+^Siglec F^+^ CD64^+^); DCs, based on expression of classic DC markers (CD45^+^CD24^+^CD64^-^CD11c^+^MHC-II^+^); or PMNs (CD45^+^MHC-II^−^GR1^+^). Similar to the BAL analysis, the percentage of neutrophils were significantly increased in *Zip8*KO mice compared to their WT counterparts (Figure 7A.) There was a trend for an increase in the total DC numbers in the lung of *Zip8*KO cecal transplant recipient mice (Figure 7B). Similarly, while not reaching significance, alveolar macrophages had decreased in mice recolonized with *Zip8*KO-microbiota (Figure 7C). Conversely, there was an increase in inflammatory macrophages in the lungs of mice recolonized with *Zip8*KO-microbiota (Figure 7D) with no change in anti-inflammatory macrophages (Figure 7E). The analysis of CD4+ (Figure 7F) and CD8+ T-cells (Figure 7G), as well as NK cells (Figure 7H), exhibited a trend toward lower cell numbers in *Zip8*KO cecal transplant recipients, although statistical significance was only achieved for CD4 T-cells. These results suggest that ZIP8 loss alters the lung immune cell landscape in response to pneumococcal infection.

### 2.6. Zip8-Mediated Dysbiosis Increases Lung and Intestinal Damage Following S. pneumoniae Infection

Mice were infected with *S. pneumoniae* (4 × 10^8^ CFUs), and the pulmonary and intestinal epithelial damage was assessed. Mice recolonized with *Zip8*KO-derived fecal samples exhibited marked increases in pulmonary and intestinal permeability, as measured by circulating levels of surfactant protein-D (SPD-1, a recognized biomarker of lung damage [29]) and the intestinal fatty-acid binding protein (IFABP, a recognized biomarker of intestinal damage [30,31]), respectively (Figure 8A,B). These results are indicative of an increase in pulmonary and intestinal permeabilities following *S. pneumoniae* infection, which could explain, in part, the increase in lung inflammation, as epithelial damage/permeability is known to increase inflammation and the bacterial burden in the lungs and spleen, as previously shown.

### 2.7. Zip8KO-Derived Microbial Products Suppress Immune Function

To understand whether microbial-derived products (metabolites and cellular components) from the gut microbiota directly influenced immune cell function, we first generated microbial products under anaerobic conditions and then incubated MH-S cells with each solution for 6 or 24 h at a concentration of either 10 or 20%, respectively (*v*/*v*). Cytotoxicity was evaluated by the LDH release, with minimal differences observed between WT and *Zip8*KO-derived microbial products (Figure 9A). As expected, microbial products isolated from either WT or *Zip8*KO intestinal communities resulted in increased IL-6 and TNFα production at 6 and 24 h; however, MH-S cells incubated with microbial products derived from *Zip8*KO cecal samples exhibited a significant decrease in the release of both cytokines when compared to microbial products isolated from WT microbiota (Figure 9B,C).

## 3. Discussion

Zinc is required for the growth and sustenance of eukaryotic and prokaryotic cells, thereby creating a “tug of war” for nutrient acquisition between the host and the microbe. Transmembrane spanning metal ion transporters serve as the primary conduit of micronutrient biodistribution, in favor of the host, to eradicate pathogens [32,33]. Transporter-mediated increases in eukaryotic cytosol metal ion content leads to the disruption of protein function in pathogens [33,34] and results in the nutrient deprivation of elements essential for bacterial growth and survival [35]. Ten Zn export proteins (ZnTs) and 14 Zn import, Zrt-Irt-like-proteins (ZIPs) control Zn homeostasis in mammals [36]. Collectively, these highly conserved transporters regulate Zn homeostasis under normal and stress-induced conditions in mammals. Our group discovered that ZIP8 is indispensable in myeloid cell-mediated host defense against pneumococcus in the lung through the alteration of Zn-dependent cell signaling networks that influence the immune response [37]. Furthermore, it has been shown in humans and in animal models that aging is a predisposing factor for Zn deficiency, which can adversely impact both innate and adaptive immunity [16,17,38,39,40]. To expand further upon these important observations, we now provide evidence that the impact of Zn and ZIP8 is not limited exclusively to the lung and involves communication between the gut microbiome and the lung. To what extent this is impacted by dietary intake and aging will require additional studies.

Revealing that human myeloid cells manipulate iron in response to infection revolutionized the fields of micronutrient homeostasis and innate immunity [41]. More recently, the discovery of human Zn transporters has accelerated our understanding of Zn homeostasis in relation to myeloid cell functions, and innate immunity in relation to infectious diseases [36]. Through these important contributions, our group has revealed that ZIP8 is essential for the regulation of the host’s response to infection [22]. We believe that our group is the first to investigate the impact of the Zn transporter-mediated regulation of the gut microbiome and its impact upon pneumococcal pneumonia.

A plethora of data has emerged demonstrating the importance of the intestinal microbiota in optimal host defense against bacterial and viral respiratory infections [42,43,44,45,46,47,48,49]. For example, germ-free mice, as well as antibiotic-treated mice, are highly susceptible to pulmonary infections with bacterial pathogens, such as *K. pneumoniae* and *S. pneumoniae* [42,43]. While several mechanistic pathways have been identified that regulate the gut-lung axis [42,43], there are still many pathways and mechanisms that are currently being identified. For example, the intestinal microbiota-mediated production of various short-chain fatty acids (SCFAs) have been shown to be important for the host’s systemic immunity [28,50,51]. More precisely, SCFAs, especially butyrate, exert broad anti-inflammatory activities by affecting immune cell migration, adhesion, and cytokine expression, as well as cellular proliferation, activation, and apoptosis through the activation of signaling pathways and the inhibition of histone deacetylase. Similarly, Sencio et al. demonstrated that a sublethal infection with influenza transiently alters the composition and fermentative activity of the gut microbiota in mice, characterized by a decrease in SCFA levels [52]. Importantly, fecal transfer experiments demonstrated that the influenza A virus (IAV)-conditioned microbiota compromised lung defenses against pneumococcal infection via changes in the SCFA acetate, as recolonized mice treated with acetate had reduced pulmonary bacterial loads [52]. Our results support these data, in that *Zip8*KO-associated intestinal dysbiosis was associated with both decreased levels of butyrate and exacerbated bacterial pneumonia, which supports the critical role of SCFA in pulmonary host defense in relation to zinc dyshomeostasis. However, the specific cell types and associated signaling pathways that are adversely impacted by alterations of butyrate metabolism remain to be identified.

## 4. Materials and Methods

### 4.1. Mice

All animals were maintained under specific pathogen-free conditions in the Animal Resource Facility at the University of Nebraska Medical Center. Food and water were provided ad libitum. The research protocol used in these studies was approved by the Institutional Animal Care and Use Committee of the University of Nebraska Medical Center. All methods involving animal care and procedures in this research protocol were performed in accordance with the NIH and Office of Laboratory Animal Welfare (OLAW) guidelines. Conditional Zip8 knockout mice, referred to as *Zip8*KO, were generated, as previously described [22,37]. Briefly, heterozygous Zip8flox-neo/+ mice were bred with ROSA26:FLPe knock-in mice (The Jackson Laboratory, Bay Harbor, ME, USA) with a ubiquitous expression of FLP1 recombinase to delete the Neo cassette adjacent to the upstream loxP site. The resulting Zip8flox/+ were mated to produce Zip8flox/flox mice. PCR and DNA sequencing confirmed the removal of the FRT-flanked sequence and verified the loxP sites flanking exon 3. Zip8flox/flox mice were crossed with myeloid cell-specific LysMcre (The Jackson Laboratory) to generate the conditional *Zip8*KO. LysMcre-mediated Zip8 deletion was confirmed in lung myeloid cells at baseline and 24 h post-LPS stimulation using a ROSA reporter (as previously described) [37]. C57BL/6J wild-type counterparts were purchased from Jackson Labs and bred for experimental procedures.

### 4.2. Microbiota Collection and Cecal Adoptive Transfer

The microbiota adoptive transfer was performed, as previously described [46]. Briefly, naïve 8- to 10-week-old female C57BL/6 mice were treated (oral gavage) with a cocktail of antibiotics (ampicillin, gentamicin, neomycin, and metronidazole (all at 0.25 mg/day), as well as vancomycin (0.125 mg/day)) daily for two weeks. The cecal content collected from age- and sex-matched wild-type or *Zip8*KO mice (see above) were homogenized and prepared for recolonization. The cecal content was weighed and homogenized (1:2 *w*/*v*) in sterile PBS. Samples were then vigorously mixed and placed on ice for 10 min to allow for organic matter to settle out. Supernatants were collected and passed through sterile 2-ply gauze to remove any large organic material and were either used immediately or stored at −80 °C. Microbiota-depleted mice were then recolonized with 200 µL of the cecal microbiota from either wild-type or *Zip8*KO donors by gavage on days 2, 5, and 8 post the antibiotic treatment. One week following the final microbiota recolonization, mice were sacrificed, and baseline immune and physiological characteristics were determined. A similar cohort of mice were infected with *S. pneumoniae* via intranasal inoculation and were sacrificed 48 hrs post-infection (see below).

### 4.3. DNA Sequencing of the 16s rRNA Gene

Sequencing was performed by the University of Nebraska Medical Center’s Genomic Core. Briefly, cecal contents were flash frozen and genomic DNA extraction was performed using the QIAamp PowerFecal Pro DNA Kit (Qiagen Valencia, CA, USA). The hypervariable V3/V4 region of the 16S rRNA gene was amplified for each of the DNA samples, beginning with 12.5 ng of DNA as per Illumina’s recommended protocol outlined in the 16S Metagenomic Sequencing Library Preparation protocol (Illumina, San Diego, CA, USA). The 16S Amplicon PCR Forward Primer was: 5′-TCGTCGGCAGCGTCAGATGTGTATAAGAGACAGCCTACGGGNGGCWGCAG-3′ and the 16S Amplicon PCR Reverse Primer was: 5′-GTCTCGTGGGCTCGGAGATGTGTATAAGAGACAGGACTACHVGGGTATCTAATCC-3′. Following the generation of the amplicons, dual indices and Illumina sequencing adapters were added using the Nextera XT Index kit (Illumina catalog number FC-131-1001). Resultant libraries were multiplexed and 300 bp paired end sequencing was performed on an Illumina MiSeq instrument using V3 chemistry as per Illumina’s recommendations. In addition, internal sequencing controls (a negative water control and a positive mock community control) were included to test for the contamination of the sequencing runs.

### 4.4. Sequence Analysis

Raw sequence data was processed using R and the R packages DADA2 v1.1.5, phyloseq v1.16.2, DESeq2 v1.20.0, PICRUSt2 v2.2.0, STAMP v2.1.3, and vegan v2.3-5 [53,54,55,56,57,58]. Sequences were truncated, denoised, chimera-filtered, and clustered into sequence variants using DADA2. Operational taxonomic units (OTU) were generated in DADA2 by the taxonomic classification of sequence variants using the SILVA reference database v132. The number of unique sequence variants in a sample (alpha diversity) was calculated using the estimate_richness function in phyloseq. The beta-diversity analysis was performed using a distance-based redundancy analysis (dbRDA) on sample-wise Bray–Curtis dissimilarity distances using phyloseq and vegan. Differentially abundant OTUs were determined via DESeq2. Inferred functional capacities were determined using PICRUSt2 and the statistical significance was determined using STAMP. Graphics were generated using ggplot2 and STAMP.

### 4.5. SCFA Quantification

A Shimadzu Nexera UPLC system equipped with two pumps (LC-30 AD) and a column oven (CTO-30AS) along with an auto-sampler (SIL-30AC) were used for analyte separation. Mass spectrometric detection was performed, utilizing a LC-MS/MS 8060 system (Shimadzu Scientific Instruments, Columbia, MD, USA), equipped with a DUIS source operated in positive electrospray ionization mode. All chromatographic separations were performed with a Waters UPLC BEH C18 analytical column (2.1 × 100 mm; 1.7 um) and an Acquity UPLC C18 guard column (Waters, Inc., Milford, MA, USA). The mobile phase consisted of 0.01% formic acid in water (mobile phase A) and acetonitrile (mobile phase A), at a total flow rate of 0.35 mL/min. The chromatographic separation was achieved using a 15 min gradient elution and eight separated SCFAs. The SCFAs were extracted from the cecum and were prepared, with quality control and calibration standards, by a simultaneous extraction/derivatization pretreatment (SEDP) procedure. Briefly, cecum samples were weighed, homogenized and vortexed, diluted with water, and centrifuged (3210× *g*). The resulting supernatant was derivatized with 3-nitrophenylhydrazine (3-NPH),N-(3-dimethylaminopropyl)-N-ethylcarbodimide hydrochloride (EDC) and pyridine. The sample was further diluted (a total dilution of 55-fold) and 5 uL was prepared for injection. The assay was linear over the concentration range of 0.01 to 100 uM and the standard and control concentrations were within 90% of the expected values.

### 4.6. The Culture, Quantification, and Instillation of Streptococcus pneumoniae

The *S. pneumoniae* strain JWV500 (D39hlpA-gfp-Cam’), a generous gift from Dr. Jan-Willem Veening (University of Lausanne, Lausanne, Switzerland), was grown to a mid-log phase, aliquoted, frozen, and stored at −80 °C until further use. For lung infection studies, bacteria were grown to a log phase in Remel Mueller Hinton Broth (Thermo Fisher Scientific, Waltham, MA, USA) and were supplemented with 32 mg/mL chloramphenicol. For the quantification of pneumococci, serial dilutions of the bacteria were plated on Remel blood agar plates (Thermo Fisher Scientific) and were incubated at 37 °C with 5% CO_2_ overnight to determine the colony forming units (CFUs). For intranasal instillations, mice were lightly anesthetized using 2% isoflurane and 1 L/min of oxygen and were instilled with 4 × 10^8^ CFU of *S. pneumoniae* in 100 µL of phosphate buffered saline (PBS), equally distributed (2 × 50 µL) between the nostrils. Mice were allowed to recover between nasal instillation doses to prevent respiratory distress. The *S. pneumoniae* dose was confirmed by serial dilutions.

### 4.7. S. pneumoniae Lung and Spleen Quantification

*S. pneumoniae* levels in the lungs of mice was measured by a real-time quantitative PCR of the *lytA* gene, using the TaqMan Fast Advanced Master Mix (Sigma Aldrich, St, Louis, MO, USA), according to the manufacturer’s specifications. The following primer sequences were used: LytA forward, 5′-ACGCAATCTAGCAGATGAAGCA-3′ and LytA Reverse, 5′-TCGTGCGTTTTAATTCCAGCT-3′. The probe (5′-TGCCGAAACGCTTGATACAGGGAG-3′) was labeled at the 5′ end with 6-carboxyfluorescein (FAM), an internal ZEN Quencher, and on the 3′ end with the Iowa Black FQ Quencher (3IABkFQ; IDT). The assays were carried out with a final 20 μL reaction volume, with 2.0 μL of sample DNA. The primers and probe were added at 200 nM concentrations. A no-template control and an *S. pneumoniae*-positive DNA control were included in every run. DNA was amplified with the 7500 Real Time PCR system (Applied Biosystems, Waltham, MA, USA) by using the following cycling parameters: 95 °C for 10 min, followed by 40 cycles of 95 °C for 15 s and 60 °C for 1 min. The pulmonary burden of *S. pneumoniae* (CFU/Lung) was calculated based on a standard curve of purified *S. penumoniae* DNA derived from a known CFU value.

### 4.8. Bronchoalveolar Lavage (BAL) Fluid Analyses

Lungs were lavaged three times with 1 mL ice-cold PBS. Total cell counts were determined using a hemocytometer and differential cell counts were determined on cytospin-prepared slides stained with Hema-3 (Thermo Fisher Scientific). Cytokine and chemokine levels were measured using commercially available ELISA kits, according to the manufacturer’s instructions (BioLegend, R&D Systems, San Diego, CA, USA).

### 4.9. Tissue Processing

Lung lobes were collected and perfused with a digestion solution containing 1× HBSS (Hyclone, GE Healthcare Lifesciences, Logan, UT, USA), 1 mg/mL collagenase D (Roche, Clinton, NJ, USA), and 20 µg/mL DNase (Roche), incubated for 30 min at 37 °C, and homogenized using gentleMACS™ Octo Dissociator (Miltenyi Biotec, Auburn, CA, USA). After enzymatic digestion and red blood cell lysis (1× RBC Lysis Buffer, Invitrogen by Thermo Fisher Scientific), samples were resuspended in a FACS rinsing buffer (1× PBS supplemented with 4% FBS and 20% sodium azide) for cell surface staining.

### 4.10. Lung Histology

Whole lungs were inflated with 10% formalin (Thermo Fisher Scientific) to preserve pulmonary architecture. Lungs were processed, paraffin embedded, sectioned (4–5 µm), and stained with hematoxylin and eosin by the UNMC Tissue Sciences Core Facility. Slides were scanned using the Ventana HT iScan (Roche Diagnostic, Mannheim Germany) and images were acquired (20×) using the Ventana Image Viewer software (Roche). Slides were reviewed and semi-quantitatively assessed (assigned a score from 0 to 5, with a higher score indicting greater inflammatory changes) by a pathologist (TAW) blinded to the treatment conditions.

### 4.11. Tissue Immunostaining

Formalin-fixed, paraffin-embedded (FFPE) lung sections were stained for in situ apoptosis detection using the Click-IT Plus TUNEL Assay Alexa 594 (Thermo Fisher) according to the manufacturer’s instructions. Images were visualized using a Zeiss Observer Z1 inverted phase contrast fluorescent microscope (Carl Zeiss Microscopy, LLC, White Plains, NY, USA) and the Alexa 594 fluorescent intensity was quantified using the Image J software (NIH).

### 4.12. Cell Culture

The murine alveolar macrophage cell line, MH-S (ATCC, Manassas, VA, USA), was maintained in standard cell culture conditions with RPMI-1640 (ATCC) and 10% FBS (Corning, Corning, NY, USA) at 37 °C in a 5% CO_2_ incubator. Then, 2 × 10^5^ cells were plated in a 24-well plate and were allowed to adhere to the plate for 24 h. Cells were washed with fresh media, then stimulated with a 10% or 20% WT or *Zip8*KO microbiota cocktail, respectively, at the indicated time points.

### 4.13. SPD-1 and IFABP ELISA

The serum was collected from all mice at the point of sacrifice using BD serum separator tubes (BD Biosciences, San Jose, CA, USA). The serum was then subject to SPD-1 and IFABP ELISAs according to the manufacturer’s specifications (R&D systems, Minneapolis, MN, USA).

### 4.14. The Generation of Microbial Products

Cecal and colonic microbiota from WT and *Zip8*KO female mice were collected, homogenized in sterile PBS (1:2, *w*/*v*), and then cultured in Gifu Anaerobic Broth (diluted 1:20) for 24 h under anaerobic conditions in a BactronEZ anaerobic chamber (Sheldon Manufacturing Inc., Cornelius, OR, USA). Microbial products (i.e., metabolites, bacterial components, and the remaining culture media constituents) were collected by the removal of live and/or whole organisms by a two-step process of centrifugation (10,000× *g* for 10 min) followed by filtration through a 0.22 μm membrane filter. Microbial products were then used immediately or stored at −80 °C for later use. Microbial products were used at 10 and 20% (*v*/*v*) in all in vitro experiments.

### 4.15. Cytotoxicity and ELISA

Lactate dehydrogenase activity from the cell-free supernatant was measured using the Cytotoxicity Detection Kit (Roche Diagnostics, Mannheim, Germany). Samples were compared to a positive control by treating cells with 2% Trition X-100 for 30 min to yield 100% cell death. The supernatant was collected and centrifuged to remove cell debris. Commercially available murine IL-6 and TNFα ELISA kits were used according to the manufacturer’s instructions (BioLegend, San Diego, CA, USA). Samples were quantified following a comparison to a standard curve with known amounts of the recombinant protein.

### 4.16. Statistics

Statistical analyses were performed using GraphPad Prism version 9.1 (GraphPad Software, La Jolla, CA) and the R package vegan. Results are shown as the mean ± standard error of the mean. A *p* < 0.05 and a false discovery rate (FDR) q-value < 0.05 were deemed significant. The sample size and the number of replicates was indicated in each respective figure legend. Statistical significance was assessed using a Mann–Whitney U test for comparisons between two groups and a one-way analysis of variance (ANOVA) with Sidak’s multiple comparison test for comparisons between three or more groups. The statistical significance of the different microbiome measurements was assessed as follows: Alpha-diversity significance was inferred using Wilcoxon tests on vectors of data with corrections for multiple comparisons via FDR. Beta-diversity significance was inferred via a permutational multivariate analysis of variance using distance matrices with corrections for multiple comparisons via FDR. Differentially abundant OTUs were determined using a negative binomial distribution model from DESeq2. Inferred functional capacities were determined using PICRUSt2, and statistical significance was determined using an ANOVA followed by the Tukey–Kramer post-hoc analysis with corrections for multiple comparisons via FDR within the STAMP platform. For cell culture experiments, data are represented as means, ± SEM, from two independent experiments.

## 5. Conclusions

A dietary Zn deficiency is common in the US and, particularly, in individuals who are prone to CAP. Likewise, a commonly occurring defective variant, the ZIP8 allele (rs13107325; A391T), ranks in the top 10 of pleiotropic single nucleotide polymorphisms (SNP) identified in genome-wide association studies and within the top 1.4% of deleterious substitutions in the human genome [25]. Given the relatively high frequency of its occurrence, we believe our investigation is highly relevant. Further studies are warranted to determine whether patients that harbor the A391T allele, and who have poor dietary Zn intake, are even more prone to infections and worse outcomes and whether either, or both, can be countered with aggressive screening approaches and Zn supplementation or other preventive dietary strategies.

## Figures and Tables

**Figure 1 ijms-23-01022-f001:**
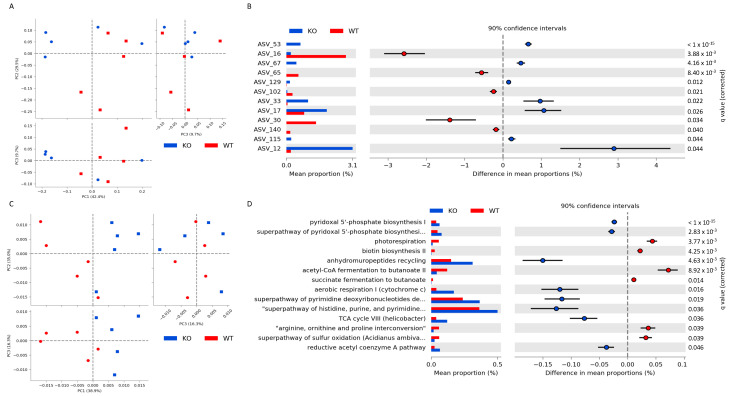
ZIP8 loss alters the intestinal microbial community and inferred functional capacity. 16S rRNA gene sequencing of WT and *Zip8*KO mice cecal microbial community. (**A**) Beta diversity of WT and *Zip8*KO mice, as determined by distance-based redundancy analysis (dbRDA) on sample-wise Bray–Curtis dissimilarity distances. (**B**) Differentially abundant ASVs as determined by DESeq2 using negative binomial generalized linear models for each taxa and Wald test for significances. (**C**) PCA plot of the inferred functional capacity of the cecal microbial communities from WT and *Zip8*KO mice. (**D**) Differentially abundant functional pathways, as determined by STAMP. *n* = 5/group.

**Figure 2 ijms-23-01022-f002:**
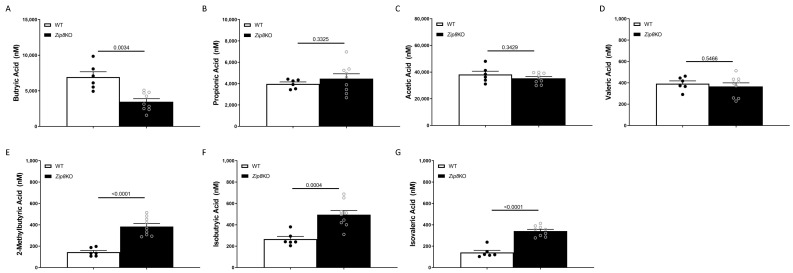
Zip8 loss decreases intestinal butyrate levels. Short chain fatty acid analysis of cecal content from WT and *Zip8*KO mice. Levels of cecal (**A**) butyric acid, (**B**) propionic acid, (**C**) acetic acid, (**D**) valeric acid, (**E**) 2-methylbutyric acid, (**F**) isobutyric acid, and (**G**) isovaleric acid. *n* = 6–9/group.

**Figure 3 ijms-23-01022-f003:**
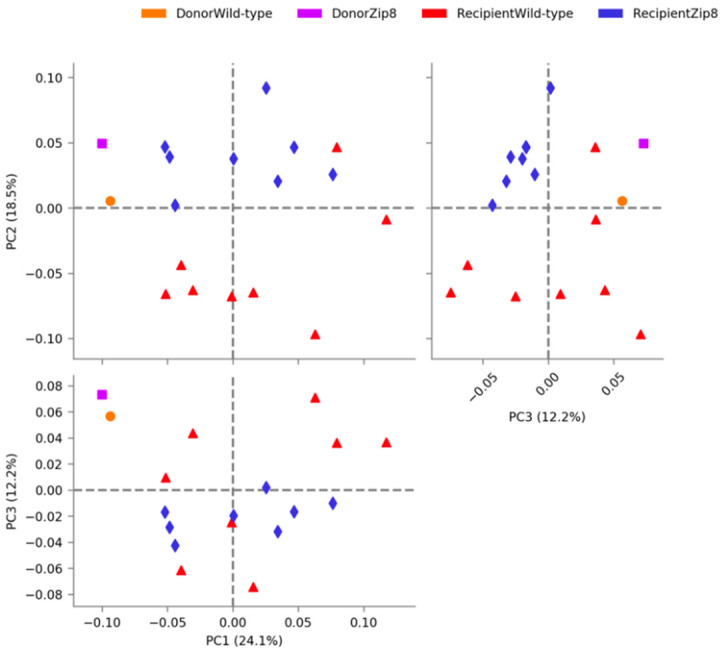
Recolonization of antibiotic cleansed mice results in maintenance of donor community structure. PCA plot of the cecal microbial communities from donor and recipient mice recolonized with the cecal microbial community from WT and *Zip8*KO mice, via STAMP. Beta diversity statistics were calculated on sample-wise Bray–Curtis dissimilarity distances. *n* = 5/group.

**Figure 4 ijms-23-01022-f004:**
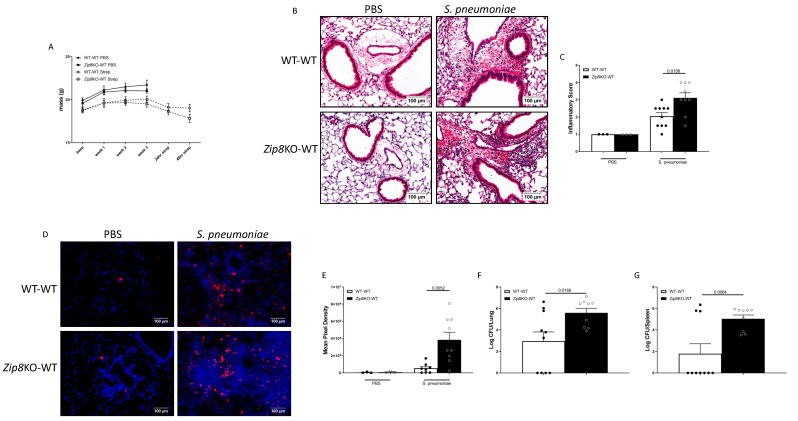
ZIP8-associated dysbiosis impairs pulmonary host defense. Mice recolonized with the cecal microbial community from WT and *Zip8*KO mice were infected with *S. pneumoniae,* and pulmonary host defense was assessed. (**A**) Weight change in recolonized mice post infection. (**B**) Representative histological staining of lung tissue, and (**C**) quantified inflammatory score. (**D**) Representative TUNEL staining of lung tissue and (**E**) quantitative mean pixel intensity. *S. pneumoniae* burden in the (**F**) lungs and (**G**) spleen. Bars represent the mean ± SEM and dots represent individual mice. *p* values are indicated in the figure and were determined by one-way ANOVA with Sidak’s multiple comparison test. *n* = 10/group, 5/experimental replicates.

**Figure 5 ijms-23-01022-f005:**
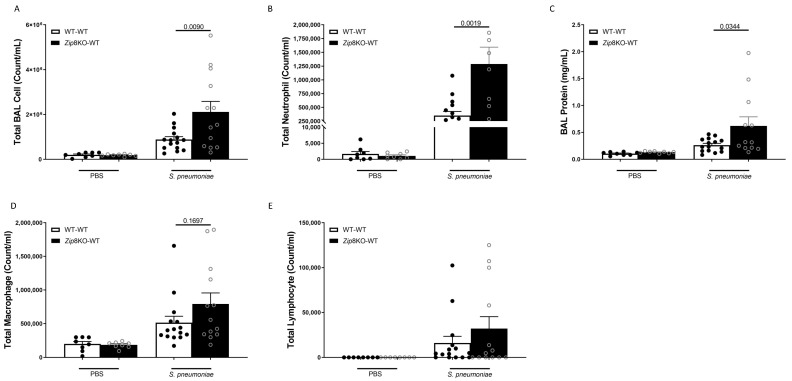
ZIP8-associated dysbiosis increases pulmonary immune cell numbers. Mice recolonized with the cecal microbial community from WT and *Zip8*KO mice were infected with *S. pneumoniae,* and the number of BAL immune cells were assessed. (**A**) Total BAL leukocytes in recolonized mice post-infection. (**B**) Total BAL neutrophils in recolonized mice post-infection. (**C**) Total BAL protein in recolonized mice post-infection. Total BAL (**D**) macrophages and (**E**) lymphocytes in recolonized mice post-infection. Bars represent the mean ± SEM and dots represent individual mice. *p* values are indicated in the figure and were determined by one-way ANOVA with Sidak’s multiple comparison test. *n* = 10/group, 5/experimental replicate.

**Figure 6 ijms-23-01022-f006:**
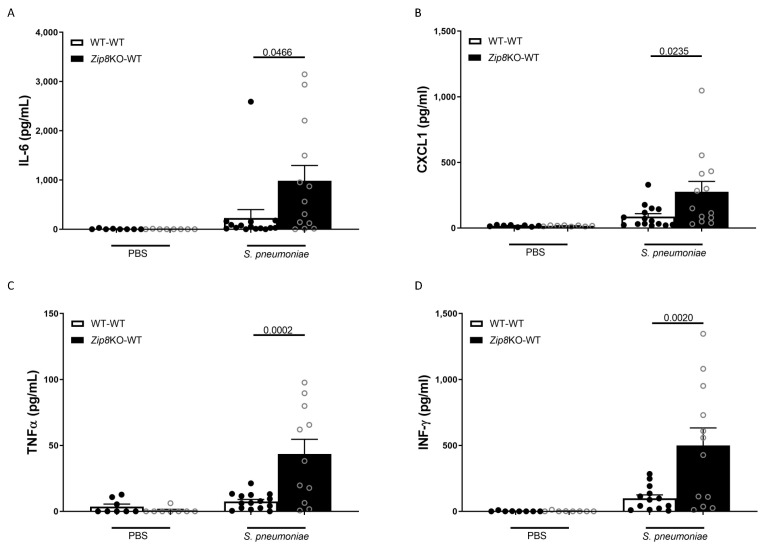
ZIP8-associated dysbiosis increases pulmonary inflammation. Mice recolonized with the cecal microbial community from WT and *Zip8*KO mice were infected with *S. pneumoniae,* and the number of pulmonary cytokine levels were assessed. (**A**) IL-6 levels in the BAL of recolonized mice post-infection. (**B**) CXCL1 levels in the BAL of recolonized mice post-infection. (**C**) TNF-α levels in the BAL of recolonized mice post-infection. (**D**) IFN-γ levels in the BAL of recolonized mice post-infection. Bars represent the mean ± SEM and dots represent individual mice. *p* values are indicated in the figure and were determined by one-way ANOVA with Sidak’s multiple comparison test. *n* = 10/group, 5/experimental replicate.

**Figure 7 ijms-23-01022-f007:**
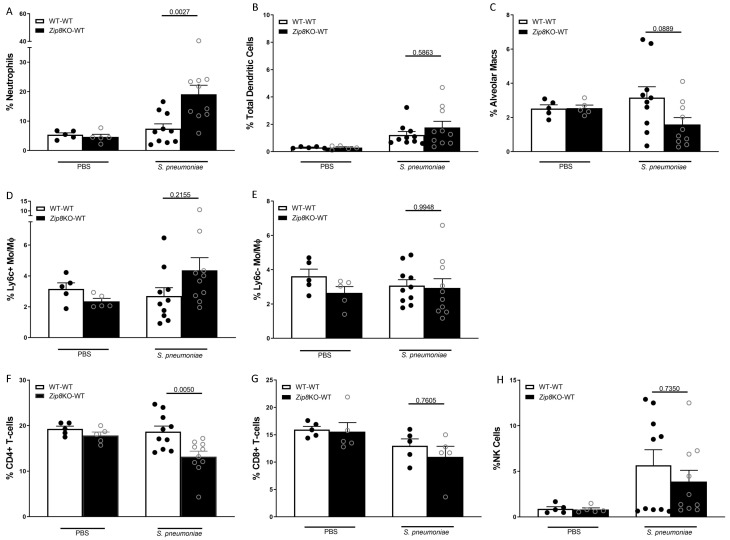
Zip8-associated dysbiosis alters the percentage of pulmonary immune cells. Mice recolonized with the cecal microbial community from WT and *Zip8*KO mice were infected with *S. pneumoniae,* and the percentage of pulmonary immune cells were assessed. Percentage of pulmonary (**A**) neutrophils, (**B**) dendritic cells, (**C**) alveolar macrophages, (**D**) inflammatory macrophages, (**E**) anti-inflammatory macrophages, (**F**) CD4+ T-cells, (**G**) CD8+ T-cells, and (**H**) NK cells. Bars represent the mean ± SEM and dots represent individual mice. *p* values are indicated in the figure and were determined by one-way ANOVA with Sidak’s multiple comparison test. *n* = 10/group, 5/experimental replicates.

**Figure 8 ijms-23-01022-f008:**
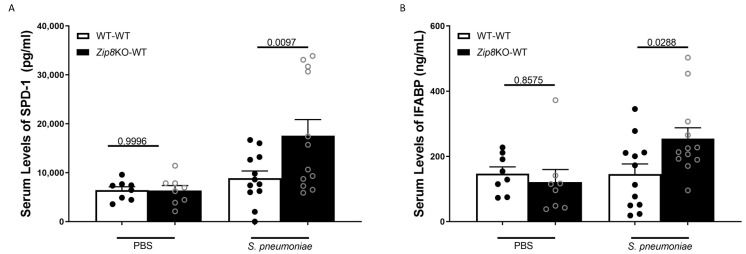
ZIP8-associated dysbiosis increases pulmonary and intestinal epithelial damage post *S. pneumoniae* infection. Mice recolonized with the cecal microbial community from WT and *Zip8*KO mice were infected with *S. pneumoniae* and the levels of circulating SPD-1 and IFABP were assessed. Circulating levels of (**A**) SPD-1 and (**B**) IFABP in mice recolonized with the cecal microbial community from WT and *Zip8*KO mice. Bars represent the mean ± SEM and dots represent individual mice. *p* values are indicated in the figure and were determined by one-way ANOVA with Sidak’s multiple comparison test. *n* = 6–12/group, 5–3/experimental replicates.

**Figure 9 ijms-23-01022-f009:**
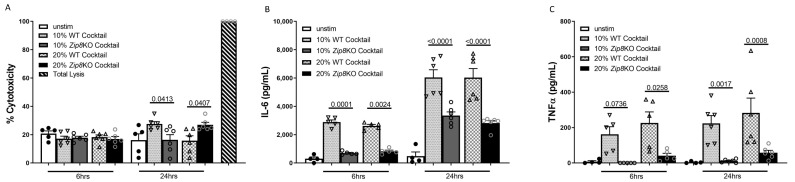
ZIP8-associated microbial products suppress macrophage cytokine production. Microbial products derived from the cecal microbial community from WT and *Zip8*KO mice were co-cultured with mouse MH-S macrophages for 6 or 24 h and macrophage viability and cytokine production was assessed. (**A**) Cytotoxicity of microbial products on MH-S cells. (**B**) IL-6 levels in the MH-S supernatant following co-culture with microbial products. (**C**) TNF-α levels in the MH-S supernatant following co-culture with microbial products. Bars represent the mean ± SEM and dots represent individual mice. *p* values are indicated in the figure and were determined by one-way ANOVA with Sidak’s multiple comparison test. *n* = 6/group, 3/experimental replicates.

## Data Availability

Sequencing data is deposited in the National Center for Biotechnology Information Sequence Read Archive (BioProject ID: PRJNA791945). All additional data, which support the findings of this study are available within the paper.

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
