# Peer review of "ZIP8-Mediated Intestinal Dysbiosis Impairs Pulmonary Host Defense against Bacterial Pneumonia"

_ijms, 2022, doi:10.3390/ijms23031022_

Round 1

Reviewer 1 Report

Dear Authors,

Thank you for this paper.

I have a comment. Some Authors have noticed parallelism in immunological changes during aging and Zn deficiency. This could lead to reduced production of thymic hormones and thymic activity overall. As a consequence, there may be a shift of the T helper cells' balance towards the T helper type 2 cells. There may also be a less effective response to vaccinations and impaired functions of the immune cells. I am wondering if age may be a factor well ( along with diet).

It is very interesting the role of ZIP8 in the maintenance of the gut microbiome. More studies on the gut-lung axis (GLA) are needed as we understand that GLA could shape the immune response and determine the outcomes of respiratory diseases.

Author Response

I have a comment. Some Authors have noticed parallelism in immunological changes during aging and Zn deficiency. This could lead to reduced production of thymic hormones and thymic activity overall. As a consequence, there may be a shift of the T helper cells' balance towards the T helper type 2 cells. There may also be a less effective response to vaccinations and impaired functions of the immune cells. I am wondering if age may be a factor well ( along with diet).

Response: Thank you for the insightful feedback.  We have added a sentence in the discussion section along with three new references that cite previous work by others that demonstrate the impact of aging and decline of Zn levels.

It is very interesting the role of ZIP8 in the maintenance of the gut microbiome. More studies on the gut-lung axis (GLA) are needed as we understand that GLA could shape the immune response and determine the outcomes of respiratory diseases.

Response: We agree that additional studies will be required in the future to understand the GLA which is a priority for us with ongoing projects between our labs. An additional sentence was added to the discussion section based on this feedback.

Reviewer 2 Report

Comments to Author:

A paper: ZIP8-Mediated Intestinal Dysbiosis Impairs Pulmonary Host 2 Defense Against Bacterial Pneumonia. The gut microbiota, 18 essential for lung immunity, is adversely impacted by a commonly occurring defective ZIP8 allele 19 in humans. Taken together they hypothesized that loss of ZIP8 function would lead to intestinal 20 dysbiosis and impair host defense against pneumonia. To test this, they utilized a novel myeloid- 21 specific Zip8KO mouse model in their studies. Comparison of the cecal microbial composition of wild 22 type and Zip8KO mice revealed significant differences in microbial community structure. Most 23 striking, upon S. pneumoniae lung infection, mice recolonized with Zip8KO-derived microbiota ex- 24 hibited increased weight loss, bacterial dissemination, and more lung inflammation compared to 25 mice recolonized with WT microbiota. Some minor comments are listed below.

  1. Define abbreviation words in the introduction and abstract sections.
  2. The English language requires significant editing (The article has many lexical and grammatical problems).
  3. The article needs a minor review.

Author Response

A paper: ZIP8-Mediated Intestinal Dysbiosis Impairs Pulmonary Host 2 Defense Against Bacterial Pneumonia. The gut microbiota, 18 essential for lung immunity, is adversely impacted by a commonly occurring defective ZIP8 allele 19 in humans. Taken together they hypothesized that loss of ZIP8 function would lead to intestinal 20 dysbiosis and impair host defense against pneumonia. To test this, they utilized a novel myeloid- 21 specific Zip8KO mouse model in their studies. Comparison of the cecal microbial composition of wild 22 type and Zip8KO mice revealed significant differences in microbial community structure. Most 23 striking, upon S. pneumoniae lung infection, mice recolonized with Zip8KO-derived microbiota ex- 24 hibited increased weight loss, bacterial dissemination, and more lung inflammation compared to 25 mice recolonized with WT microbiota. Some minor comments are listed below.

  1. Define abbreviation words in the introduction and abstract sections.

Response: Abbreviations have been put in place.

  1. The English language requires significant editing (The article has many lexical and grammatical problems).

Response: The manuscript was reviewed prior to submission by the lead, corresponding, and co-authors, all of which speak English as their primary language.  Attention was placed upon lexic and grammatic content, so we were surprised by this feedback.  Nevertheless, we had a non-biased, staff associate that works in our grant office conduct an additional read. A few minor corrections were identified and corrected.

  1. The article needs a minor review.

As previously stated above.